# Examining the Link between Isokinetic Strength Metrics and Ball Speed in Women's Soccer

Cengiz Ölmez [1], Nadhir Hammami [2], Büşra Yücelsoy [1], Soukaina Hattabi [2,3], Pedro Forte [3,4,5,*], Andrew Sortwell [4,6], Mehrzia Amani Khezami [7,8] and Alparslan İnce [1]

[1] Physical Education and Sport Department, Sport Sciences Faculty, Ordu University, Ordu 52200, Türkiye; cengizolmez@odu.edu.tr (C.Ö.); aince@odu.edu.tr (A.İ)

[2] Research Unit "Sport Sciences, Health and Movement", High Institute of Sport and Physical Education of Kef, University of Jendouba, Le Kef 7100, Tunisia; nadhir.hammami@issepkef.u-jendouba.tn (N.H.); hattabisoukaina@gmail.com (S.H.)

[3] CI-ISCE, Higher Institute of Educational Sciences of the Douro, 4560-547 Penafiel, Portugal

[4] Research Center in Sports, Health and Human Development, 6201-001 Covilhã, Portugal; sortwellandrew@gmail.com

[5] Department of Sports, Instituto Politécnico de Bragança, 5300-252 Bragança, Portugal

[6] School of Health Sciences and Physiotherapy, University of Notre Dame Australia, Fremantle, WA 6160, Australia

[7] Department of Physical and Rehabilitation Medicine, The National Institute of Orthopedics Mohamed KASSAB, La Manouba 2010, Tunisia; mehrziaamani.khezami@fmt.utm.tn

[8] Faculty of Medicine of Tunis, University of Tunis El Manar, Tunis 1068, Tunisia

[*] Correspondence: pedromiguelforte@gmail.com or pedromiguel.forte@iscedouro.pt

**Abstract:** The shot performance of female soccer players is one of the most critical factors in winning a soccer match. It is essential to thoroughly clarify the kinetic factors that can improve shot performance. This study explores the connections between ball velocity post-shooting and isokinetic knee extension (EXT) and flexion (FLX) strength performances among female soccer players. Thirteen voluntary players from professional leagues took part in the research study. The study analyzed the average and peak concentric (Con) and eccentric (Ecc) torques, isometric (Iso) strength performances at angular velocities of $60°/s$, $180°/s$ and $300°/s$, the time required to reach peak torque, and ball velocities during shooting. The relationships among these variables were investigated separately for the dominant (D) and non-dominant (ND) legs. The analysis unveiled significant correlations between ball velocities and D-EXT (Absolute) peak torque at an angular velocity of $60°/s$ ($r = 0.597$; $p < 0.05$), D-%IPS ($r = -0.580$; $p < 0.05$), and ND-FLX (Absolute) average torque ($r = 0.559$; $p < 0.05$). Moreover, notable associations were observed between ball velocities and ND-EXT (Absolute) ($r = 0.581$; $p < 0.05$), as well as ND-FLX (Absolute) ($r = 0.602$; $p < 0.05$) average torques at an angular velocity of $180°/s$. Additionally, significant relationships were found between ball velocities and peak ($r = 0.664$; $p = 0.013$) and average ($r = 0.660$; $p = 0.014$) torques generated during ND-EXT (Absolute) at an angular velocity of $300°/s$. However, the connections between the time to reach peak torque, eccentric and isometric forces, and ball velocities were not statistically significant ($p > 0.05$). The results of the study indicate that enhancing concentric isokinetic strength development at $60°/s$, $180°/s$, and $300°/s$ angular velocities, along with balanced strengthening of the ND extremity, holds paramount importance in elevating shot performance among female soccer players, particularly in the context of rapid shot strategies.

**Keywords:** female soccer players; shot velocity; isokinetic strength; isometric contraction; eccentric contraction; concentric contraction

## 1. Introduction

Soccer is a sport that demands not only intricate techniques and tactics but also high levels of physical performance [1–3]. Strength and power serve as crucial benchmarks for

assessing physical performance [4]. Isokinetic dynamometers are widely recognized as the gold standard for measuring both strength and power [5–7], making them a common choice in clinical research, particularly in studies related to various health-related injuries and conditions [8,9]. Therefore, it is crucial to accurately determine the association between sport-specific skills and iso-kinetic force parameters to determine the need for training sports skills using optimal isokinetic exercise protocols. Effectively performing a shot (i.e., kick at the goal) is a critical aspect of soccer for achieving results, and it should be trained using the most suitable training methods [10,11]. To enhance shot performance, understanding the body biomechanics during a shot and the relationship with the explosive strengthening of relevant extremities and body regions is critical [12]. This approach can improve shot performance, increasing goal success rates and overall match victory probabilities.

Effective shot performance by soccer players can significantly affect match outcomes [13,14]. The amount of force applied to the ball in conjunction with accuracy plays a decisive role in the game, facilitating the creation of goal scoring opportunities that can affect the course of a match [15]. From a biomechanical standpoint, correct ball striking combined with increased applied force accelerates the ball at greater speed.

Existing research suggests that isokinetic tests at moderate and high angular velocities are sufficient for monitoring strength training programs in soccer [16]. However, the contradiction arises from that while torque and ball speed in soccer are often positively correlated, only a few cases have statistically significant relationships between them [17]. In contrast, some studies report inconsistent relationships between strength and ball striking speed in soccer [18]. Therefore, the question arises whether isokinetic torques can predict ball speed, a significant determinant of players' overall performance, given that isokinetic tests are considered the gold standard for measuring strength. Thus, the most suitable measurement models for shot performance in soccer have not been fully understood. Consequently, a thorough classification of isokinetic force variables is required. The literature review we conducted showed that limited studies have focused on concentric contractions in the relation between ball speed and knee isokinetic strength. No study has combined both concentric and eccentric contractions, and in this regard, our study aims to fill this significant gap.

Studies highlight the reliability of speed data obtained with handheld radar devices, which are frequently utilized in biomechanical research for measuring the velocity of objects [19–21]. Consequently, the speed at which the ball travels during a shot can be reliably and easily gauged using such equipment [22,23]. However, one of the primary focuses should also revolve around gaining a comprehensive understanding of the kinetics of ball speed. This understanding can serve as the basis for creating training protocols aimed at enhancing ball speed. Considering that power is the product of strength and speed, it becomes apparent that a greater increase in both speed and strength will lead to superior power augmentation [24,25]. Moreover, given the positive relationship between speed and strength production in power generation, it becomes imperative to conduct a detailed examination of concentric, eccentric, and isometric forces that could influence ball speed.

This research study aims to identify the potential relationships between isometric-, concentric-, and eccentric-contraction-based isokinetic forces and ball velocities generated during shooting in female soccer players competing at a professional level. We hypothesized that there are significant relationships between concentric contraction torques and shooting performances at low, medium, and high angular velocities.

## 2. Materials and Methods

### 2.1. Participants

The study was conducted with the voluntary participation of 13 elite-level female soccer players. The sample size was determined using G*power (version 3.1.9.6, Kiel, Germany) analysis ($p = 0.6$; $\alpha = 0.05$; $1 - \beta = 0.8$) [26]. The research was carried out at the

beginning of the season. The players who participated are part of professional teams in the nation's second league and regularly train for approximately 10 h a week over five days. Detailed information regarding the players' demographic and physical characteristics can be found in Table 1.

**Table 1.** Physical and demographic characteristics of the players.

|  | X ± SD | Range |
|---|---|---|
| Age (yrs.) | 17.77 ± 1.74 | 16.00–23.00 |
| TSE (yrs.) | 4.50 ± 2.00 | 2.00–8.00 |
| TPSE (yrs.) | 2.46 ± 1.25 | 0.33–5.00 |
| BW (kg) | 51.72 ± 4.33 | 45.30–58.20 |
| Ht (cm) | 160.92 ± 4.8 | 152.00–174.00 |
| BMI (kg/m$^2$) | 19.96 ± 1.33 | 17.70–21.99 |
| LBM (%) | 40.62 ± 2.9 | 36.8.-45.8. |
| SLM (%) | 37.65 ± 2.68 | 34.00–42.50 |
| WHR (%) | 0.71 ± 0.01 | 0.68–0.72 |

TSE: total soccer experience; TPSE: total professional soccer experience; BW: body weight; Ht: height; BMI: body mass index; LBM: lean body mass; SLM: soft lean mass; WHR: waist-to-hip ratio.

Participation criteria for the study included being female, being at least 15 years old (minimum age limit for Women's A category), not having undergone gender transition, not having any physical or psychological issues that could hinder participation, not undergoing hormonal treatment, actively and officially playing in at least the second league, and being willing to participate voluntarily. Players who did not meet these criteria were excluded from the study.

All players were provided with information about the study, including potential benefits and risks. After a verbal explanation, written informed consent forms (prepared in accordance with the Helsinki Declaration) were given to all players. For players under 18 years of age, their parents were also informed about the study, and written informed consent was obtained from them. The study was conducted in accordance with the ethical principles of the European Convention and the Helsinki Declaration [27] and was approved by the University Clinical Research Ethics Committee (No: 2023-137/O.U).

### 2.2. Design and Procedures

The study was completed in four consecutive phases, as depicted in Figure 1. In the first phase, procedures such as recording players' demographic information, informing them about the study, and verifying their eligibility were carried out. Additionally, during this phase, the players' dominant foot was determined. To identify their dominant foot, the players were assigned tasks such as kicking a ball, picking up a marble, tracing shapes, and stamping out a simulated fire with their feet. The foot that the players preferred or were more competent at using during these tasks was considered their dominant foot [28].

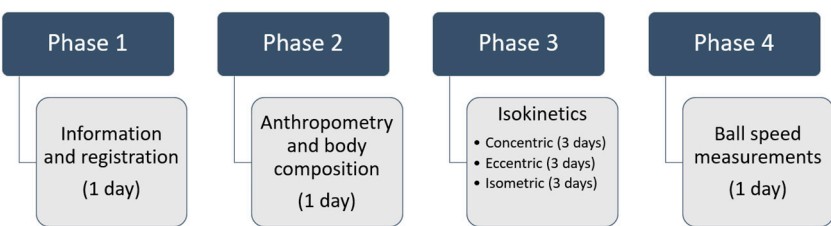

**Figure 1.** Flow chart of the research.

On the second day of the data collection phase, players' body composition, body weight, height, and body mass index (BMI) values were determined. Additionally, players were provided with information and instructions about the isokinetic force measurement device, the tests, and measurement protocols by an assistant. This sharing of information

aimed to ensure players' understanding of the isokinetic dynamometer and the measurements, as well as their mental preparedness. On the second day of the study, players were reminded not to engage in strenuous activities and to arrive for the isokinetic measurements in a well-rested state.

The third phase involved the isokinetic measurements, which consisted of consecutive phases of concentric (Con), eccentric (Ecc), and isometric (Iso) contractions. Players were divided into three different groups (5, 4, and 4 players, respectively) for each isokinetic phase to mitigate fatigue-related performance losses and enable testing at similar time intervals. The participants visited the laboratory on consecutive days, and the isokinetic tests began as previously recommended (around 16:00 and approximately 3 h after a standard meal) [29].

In the fourth phase, all players were asked to visit the research facility on the same day, and ball speeds during penalty kicks were examined. Penalty kicks were initiated at the same time that previous testing occurred.

*2.3. Data Collection*

2.3.1. Anthropometrics and Body Composition

The players' body weight (BW), lean body mass (LBM), soft lean mass (SLM), and waist-to-hip ratio (WHR) values were measured using a body composition analysis device (Jawon Body Composition Analyzer Model X-Scanplus II, Seoul, Korea) [30]. Measurements were taken in a barefoot anatomical standing position. The consumption of caffeine and alcohol was prohibited within the 24 h preceding the measurements. The measurements were conducted at 10:00 a.m. before breakfast and after the players had addressed their bathroom needs. Additionally, considering a factor that could influence body composition, players were not in their menstruation period [31–33].

The players' height measurements were taken using a stadiometer (Holtain Ltd. Crymych, UK). Players were asked to stand in an anatomical position with bare feet and heels together. The point where the upper table of the stadiometer touched the head was recorded in centimeters.

The players' body mass index (BMI) values were calculated using their heights and body weights ($BW/Ht^2$) [34].

2.3.2. Isokinetic Strength Evaluation of Muscular Performance

Players' isokinetic performances were measured using an isokinetic dynamometer of Humac NORM type (CSMI, Stoughton, MA, USA) [35–37]. Isokinetic measurements of both the dominant (D) and non-dominant (ND) extremities were used to assess peak and average torque for each joint. Data were obtained for absolute torque (Abs), torque normalized to body weight (%BW), and the ratio of ipsilateral torque of agonist–antagonist muscles (%IPS). Calculations were made when the tested limb was parallel to the ground, accounting for the effect of gravity. The study utilized the Con/Con, Ecc/Ecc, and Iso protocols. All tests were conducted by the same laboratory assistant, who received training in the Humac NORM test.

For each joint, a warm-up set consisting of five repetitions at 50% perceived effort was performed prior to the test [38]. The purpose of the warm-up set was to familiarize the subjects with the isokinetic testing sensation and provide a specific warm-up for the muscles to be tested [39]. Following the warm-up and between sets, players were given a 90 s rest period. The laboratory assistant verbally encouraged the tested athlete throughout each test session. Concentric measurements were conducted at angular velocities of 60–180–300°/s, as previously recommended for young soccer players [40–42]. Eccentric measurements were performed at an angular velocity of 60°/s. As this test speed has predominantly been adopted in the scientific literature and is considered the optimal testing method for assessing peak torque and maximum power, there is consensus on performing a 5-repetition test at 60°/s isokinetic speed [42]. Finally, there is agreement in the literature that an isokinetic protocol should encompass three speeds, as the protocol

should provide information on maximum strength, muscle endurance capacity, and the speed–force relationship [42]. Hence, in our study, 5 repetitions were conducted for both concentric and eccentric contractions.

The isokinetic knee EXT-FLX test (con/con and ecc/ecc) was conducted with the athlete seated. Prior to the test, the isokinetic device's seat back angle was adjusted to 85 degrees, and the seat rotation angle was set at 40 degrees. The dynamometer's rotation angle was also set to 40 degrees, while the tilt angle remained at 0 degrees and the height at eight units. The monorail, back translation, and fore–aft values were fine-tuned to precisely align the athlete's knee cap and the dynamometer shaft. The athlete's stationary leg and torso were securely fastened using the dynamometer straps. The athlete's leg was set to a zero-degree angle at full extension, and a 90-degree flexion angle was designated, resulting in a movement range spanning from 0 to 90 degrees. For the isometric measurements, the leg angle was 45 degrees, and the contraction duration was 5 s. Only EXT strength was measured in isometric contractions [43].

### 2.3.3. Ball Speed

The ball velocities of the players were measured using a highly reliable hand-held radar gun (Bushnell 101911, Overland Park, KS, USA) that has previously been reported to have a strong correlation (r = 0.88) [19–21]. Before the measurements, the players engaged in a 15 min warm-up session with the ball, followed by additional warm-up exercises consisting of taking ten penalty kicks, each from a distance of 11 m. Subsequently, they were instructed to take shots at a goal with dimensions of 2 m × 3 m from a distance of 11 m. The radar device was manually placed behind the goal, precisely at its center, to record the shot with the highest ball speed among three attempts, measured in kilometers per hour (km/h).

### *2.4. Data Analysis*

Statistical analyses were performed using SPSS (version 25.0, Chicago, IL, USA). The results are reported as mean values ± standard deviation (X ± SD) and minimum–maximum (Range). Data normality and homogeneity of variances were assessed using the Shapiro–Wilk test, Q–Q plots, and Levene's test, respectively. The relationships between players' ball speed and isokinetic strength parameters were analyzed using Spearman correlation analysis. In correlation analysis, the correlations were categorized as weak (>0.30), moderate (0.3–0.5), high (0.6–0.8), and excellent (0.8–1.0) [44]. The significance level was set at $p \leq 0.05$.

### 3. Results

The examination revealed that the average ball speed generated by the players after penalty kicks was 67.38 ± 15.81 km/h (range: 34–82 km/h). Players' ball speeds are presented in Figure 2.

The players' isokinetic concentric torques are provided in Table 2, eccentric torques in Table 3, isometric torques in Table 4, and time to reach peak torque in Table 5. When examining the relationships between players' ball speeds and isokinetic strength performances (Figure 3), it was found that the relationships were only significant in absolute torques (Abs) and concentric contractions ($p < 0.05$). The relationships between eccentric and isometric strengths and ball speeds were not significant ($p > 0.05$). Therefore, when investigating the peak and average isokinetic strength performances at 60°/s angular velocity, significant moderate correlations were observed between peak torque parameters, D-EXT (r = 0.597; $p = 0.031$), and D-%IPS (r = −0.580; $p = 0.038$), and ball speeds. Similarly, significant moderate correlations were observed between average torque parameters, ND-FLX, and ball speeds (r = 0.559; $p = 0.047$). When examining the relationships between players' average isokinetic strength at 180°/s angular velocity and ball speeds, it was observed that there were significant moderate to high correlations between ball speeds and ND-EXT (r = 0.581; $p = 0.037$), as well as between ball speeds and ND-FLX (r = 0.602; $p = 0.029$). It was observed

that there existed significant positive correlations at a high level between ball speeds and peak torque values (r = 0.664; *p* = 0.013), as well as between ball speeds and average torque values (r = 0.660; *p* = 0.014), both produced during ND-EXT at an angular velocity of 300°/s.

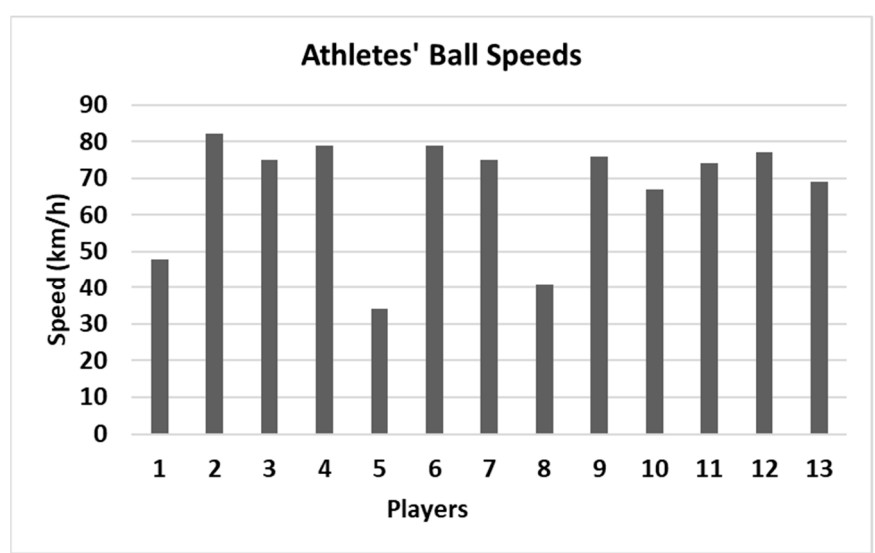

**Figure 2.** Players' ball speeds.

**Table 2.** Concentric FLX and EXT strengths obtained by players.

| | | Peak Torque X ± SD | Range | Average Torque X ± SD | Range |
|---|---|---|---|---|---|
| 60°/s (NM) | D-EXT (Abs) | 129.54 ± 18.42 | 88–160 | 86.46 ± 12.21 | 59–106 |
| | D-EXT (%BW) | 253.15 ± 35.05 | 185–310 | 169.38 ± 24.06 | 123–202 |
| | D-FLX (Abs) | 62.85 ± 9.62 | 45–75 | 47.92 ± 7.96 | 32–60 |
| | D-FLX (%BW) | 122.46 ± 21.21 | 83–167 | 93.92 ± 16.93 | 66–134 |
| | D-%IPS | 48.46 ± 4.29 | 42–55 | 55.69 ± 7.19 | 47–70 |
| | ND-EXT (Abs) | 126.54 ± 19.26 | 87–155 | 83.92 ± 12.51 | 58–103 |
| | ND-EXT (%BW) | 242.38 ± 39.38 | 132–295 | 163.77 ± 20.18 | 121–198 |
| | ND-FLX (Abs) | 60.08 ± 9.28 | 37–75 | 45 ± 6.46 | 29–56 |
| | ND-FLX (%BW) | 117.15 ± 17.52 | 77–140 | 88.31 ± 13.17 | 62–114 |
| | ND-%IPS | 47.69 ± 6.54 | 41–64 | 54.15 ± 8.16 | 47–78 |
| 180°/s (NM) | D-EXT (Abs) | 86.15 ± 12.17 | 56–100 | 150.00 ± 25.30 | 96–197 |
| | D-EXT (%BW) | 168.38 ± 25.62 | 116–224 | 293.69 ± 56.1 | 200–437 |
| | D-FLX (Abs) | 43.46 ± 10.17 | 19–56 | 79.08 ± 21.29 | 34–117 |
| | D-FLX (%BW) | 84.69 ± 19.96 | 39–122 | 154.46 ± 44.33 | 70–259 |
| | D-%IPS | 49.85 ± 7.68 | 34–59 | 52.08 ± 9.1 | 35–65 |
| | ND-EXT (Abs) | 86.62 ± 13.78 | 56–108 | 150.46 ± 22.51 | 95–183 |
| | ND-EXT (%BW) | 168.62 ± 21.55 | 116–194 | 293.38 ± 38.59 | 198–338 |
| | ND-FLX (Abs) | 42.31 ± 7.11 | 24–54 | 76.15 ± 12.77 | 42–94 |
| | ND-FLX (%BW) | 82.31 ± 12.87 | 51–101 | 149.00 ± 24.73 | 88–191 |
| | ND-%IPS | 49.15 ± 4.71 | 44–57 | 50.62 ± 4.87 | 44–57 |
| 300°/s (NM) | D-EXT (Abs) | 63.00 ± 11.25 | 39–79 | 143.00 ± 33.87 | 92–215 |
| | D-EXT (%BW) | 123.15 ± 22.93 | 83–176 | 280.31 ± 69.51 | 193–426 |
| | D-FLX (Abs) | 31.46 ± 10.41 | 4–42 | 65.69 ± 23.28 | 7–98 |
| | D-FLX (%BW) | 61.69 ± 19.79 | 9–86 | 128.15 ± 46.91 | 15–218 |
| | D-%IPS | 49.15 ± 14.61 | 10–65 | 45.46 ± 13.79 | 8–59 |
| | ND-EXT (Abs) | 63.23 ± 10.51 | 41–83 | 138.62 ± 25.92 | 86–182 |
| | ND-EXT (%BW) | 122.92 ± 18.18 | 86–161 | 270.92 ± 50.03 | 180–393 |
| | ND-FLX (Abs) | 29.00 ± 8.67 | 12–41 | 60.92 ± 22.15 | 21–98 |
| | ND-FLX (%BW) | 57.38 ± 16.57 | 27–83 | 119.62 ± 45.59 | 42–218 |
| | ND- %IPS | 46.00 ± 10.61 | 23–57 | 42.85 ± 11.36 | 18–56 |

X: arithmetic mean; SD: standard deviation; D: dominant; ND: nondominant; EXT: extension; FLX: flexion; Abs: absolute value; BW: body weight; IPS: ipsilateral ratio.

**Table 3.** Eccentric contraction-derived FLX and EXT strengths of the players.

| | Peak Torque | | Average Torque | |
| --- | --- | --- | --- | --- |
| | X ± SD | Range | X ± SD | Range |
| D-EXT (Abs) | 139.46 ± 34.23 | 92–184 | 91.54 ± 21.49 | 57–123 |
| D-EXT (%BW) | 271.31 ± 58.02 | 167–349 | 177.85 ± 36.48 | 121–224 |
| D-FLX (Abs) | 68.92 ± 17.63 | 39–103 | 51.62 ± 14.68 | 29–79 |
| D-FLX (%BW) | 135.69 ± 35.70 | 72–191 | 101.00 ± 28.85 | 53–147 |
| D-%IPS | 50.77 ± 13.26 | 37–86 | 57.08 ± 14.75 | 42–97 |
| ND-EXT (Abs) | 141.54 ± 40.18 | 69–199 | 91.92 ± 24.78 | 49–131 |
| ND-EXT (%BW) | 275.46 ± 73.84 | 155–390 | 179.00 ± 46.38 | 103–257 |
| ND-FLX (Abs) | 65.85 ± 18.02 | 37–94 | 46.15 ± 14.69 | 23–74 |
| ND-FLX (%BW) | 128.77 ± 35.84 | 66–185 | 90.31 ± 29.08 | 42–136 |
| ND-%IPS | 49.31 ± 21.82 | 37–120 | 52.46 ± 22.95 | 31–124 |

X: arithmetic mean; SD: standard deviation; D: dominant; ND: nondominant; EXT: extension; FLX: flexion; Abs: absolute value; BW: body weight; IPS: ipsilateral ratio.

**Table 4.** Isometric contraction-derived EXT strengths of the players.

| | Peak Torque | | Average Torque | |
| --- | --- | --- | --- | --- |
| | X ± SD | Range | X ± SD | Range |
| D (Abs) | 156.08 ± 33.48 | 115–243 | 132.92 ± 26.88 | 99–202 |
| D (%BW) | 304.69 ± 61.44 | 218–468 | 260.08 ± 51.55 | 188–390 |
| ND (Abs) | 151.85 ± 37.48 | 107–221 | 128.85 ± 35.51 | 83–195 |
| ND (%BW) | 294.69 ± 63.26 | 218–426 | 249.23 ± 59.45 | 173–373 |

X: arithmetic mean; SD: standard deviation; D: dominant; ND: nondominant; Abs: absolute value; BW: body weight.

**Table 5.** Players' time to reach peak torque.

| | | Peak Torque Time | |
| --- | --- | --- | --- |
| | | X ± SD | Range |
| 60°/s Con/Con | D-EXT | 0.52 ± 0.09 | 0.42–0.69 |
| | D-FLX | 0.67 ± 0.19 | 0.41–1.19 |
| | ND-EXT | 0.54 ± 0.12 | 0.37–0.77 |
| | ND-FLX | 0.59 ± 0.15 | 0.41–0.93 |
| 180°/s Con/Con | D-EXT | 0.26 ± 0.02 | 0.23–0.31 |
| | D-FLX | 0.26 ± 0.08 | 0.12–0.44 |
| | ND-EXT | 0.27 ± 0.03 | 0.23–0.32 |
| | ND-FLX | 0.27 ± 0.04 | 0.21–0.35 |
| 300°/s Con/Con | D-EXT | 0.19 ± 0.02 | 0.14–0.22 |
| | D-FLX | 0.18 ± 0.05 | 0.03–0.24 |
| | ND-EXT | 0.19 ± 0.01 | 0.17–0.22 |
| | ND-FLX | 0.18 ± 0.03 | 0.13–0.22 |
| 60°/s Ecc/Ecc | D-EXT | 0.97 ± 0.23 | 0.40–1.35 |
| | D-FLX | 1.08 ± 0.30 | 0.56–1.46 |
| | ND-EXT | 0.99 ± 0.21 | 0.71–1.34 |
| | ND-FLX | 0.99 ± 0.44 | 0.34–1.50 |
| Iso Ecc | D | 4.31 ± 1.14 | 0.82–5.00 |
| | ND | 3.00 ± 1.25 | 0.87–4.81 |

X: arithmetic mean; SD: standard deviation; D: dominant; ND: nondominant; EXT: extension; FLX: flexion.

The relationships between the rates of reaching peak torque in the concentric, eccentric, and isometric contractions at 60–180–300°/s angular velocities and ball speeds were found to be statistically non-significant ($p > 0.05$).

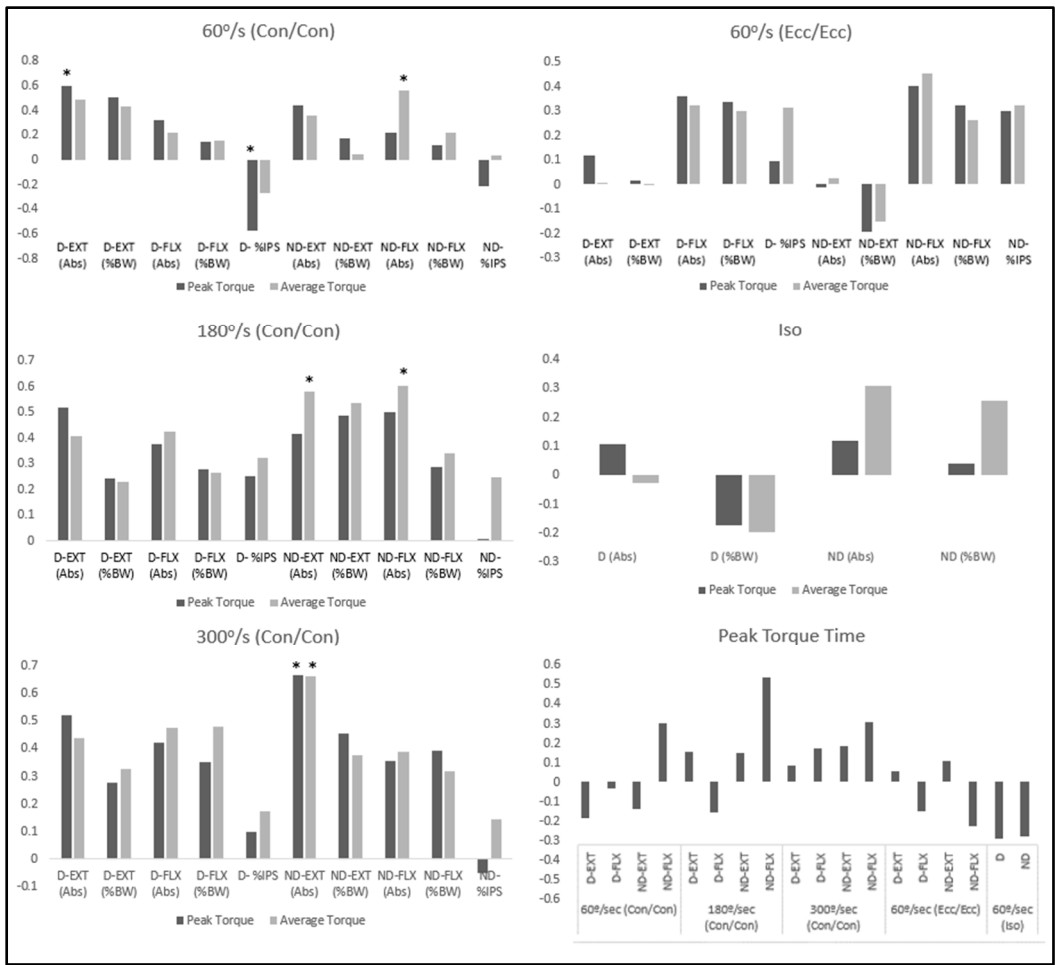

**Figure 3.** Relationships between concentric, eccentric, and isometric strength; time to peak torque; and ball speed. * $p < 0.05$.

## 4. Discussion

The aim of this study was to determine the isokinetic strength parameters associated with shooting speed in soccer. For this purpose, the relationships between the concentric, eccentric, and isometric torques produced by elite-level female soccer players and their time to reach peak torque and the ball velocities generated during shooting were examined.

The conducted analysis found that the relationships between ball velocity and isokinetic strength occurred only during concentric contractions. Accordingly, the isokinetic torques produced by players' dominant side and at 60°/s angular velocity during the EXT phase were moderately and positively correlated with the ball velocities. When analyzing the biomechanics of shooting, previous reports have indicated no relationship between the angle of approach to the ball and shot accuracy or ball velocity [45]. In soccer, the striking action progresses from proximal to distal, with the extension strength generated at the knee during the final phase creating a propulsive force on the ball [46]. Therefore, the force generated through whole-body dynamics is transferred to the ball at this stage. The obtained result suggests that an increase in knee EXT torque at 60°/s angular velocity could enhance shot velocity.

In soccer, game conditions can influence kicking technique, with the dominant leg being more affected due to factors such as a greater training history and slower adaptation to different conditions compared to the non-dominant leg [47]. As a result, the dominant leg is often preferred for kicks [48,49]. Although the kick is executed primarily with the dominant foot, the contribution of the non-dominant foot is significant due to its role in maintaining balance [48]. The findings demonstrated that the non-dominant extremity's

torque production during EXT and FLX at 60°/s and 180°/s, and during EXT at 300°/s angular velocities, was associated with ball speed. This result indicates that strengthening both the EXT and FLX strengths of the non-dominant extremity during a kick could positively impact kick performance. This is because the non-dominant extremity provides essential support for the dominant foot's healthy, balanced, and explosive impact on the ball during a kick. Additionally, as the only extremity in contact with the ground during a kick, the non-dominant extremity is responsible for overall body balance [48,49]. A balanced stance is crucial for transferring force effectively from the body to the foot and from the foot to the ball [46], and the non-dominant lower extremity contributes to maintaining this balance.

Previous studies have emphasized that players' EXT-FLX strength ratios should fall within the 50–80% range to mitigate the risk of imbalanced strength distribution and potential injuries [50–54]. The examination revealed that a decreased EXT-FLX ratio in the dominant extremity positively influenced ball speed. However, no significant relationship between this ratio and ball speed was found in the non-dominant extremity. Accordingly, while increased EXT strength in the foot used for kicking is crucial, the balanced development of both EXT and FLX strengths in the non-dominant foot is equally important. Considering the dynamic nature of modern soccer and the potential utilization of both feet for kicking, balanced strengthening of both extremities is advisable.

The relationships between non-dominant (ND) EXT peak and average torque at 300°/s angular velocity and ball speeds were determined to be highly significant and positively correlated. This finding stands as the most robust evidence influencing ball speed since the relationships between both average and peak isokinetic strength and ball speed are of high significance. However, the relationships between strength generated in the dominant direction and ball speeds were not significant. Previous reports had indicated a decrease in maximal torque at increased angular velocities [55]. Nevertheless, the 300°/s angular velocity tests the presence of explosive force at high speeds [42]. The outcome suggests that the maximum explosive force generated by a non-dominant extremity possessing a balanced agonist–antagonist ratio positively supports dominant extremity performance. This is significant since the final phase of force transmission is the contact phase with the ball, where the performance of the extremity remaining on the ground significantly affects kick performance. This study's findings underscore the importance of strong and balanced strength development in the non-dominant extremity.

The relationships between isometric and eccentric contractions of players' generated strength and ball speed were found to be non-significant. Previous studies have reported high strength torque production during isometric and eccentric contractions [56]. Additionally, it has been noted that immediately after isometric, concentric, and eccentric exercises with the same torque–time integral, similar neuromuscular fatigue is induced. However, the force loss after isometric and eccentric exercises lasts longer than after concentric exercises, and changes in peripheral fatigue parameters are most significant after eccentric exercises [57]. These results indicate the advantages of isometric and eccentric exercises in terms of generated torque. However, the findings of our research study show that these high torques do not influence ball speed. Considering the importance of force transmission for athletic performance, it can be concluded that concentric exercises may be more beneficial for soccer. This is because in soccer, kicks are executed using concentric contractions of the lower extremities. Therefore, it is natural for the force produced through concentric contractions to be more dominant during ball kicks. Moreover, the findings suggest that force exercises tailored to the specific sport and related movements can be more usable and suitable for enhancing athletic performance.

The relationships between players' time to peak torque during isokinetic strength measurement and ball speed were insignificant. This finding indicates that there is no significant association between the time to reach peak torque on the isokinetic dynamometer and the force generated during ball kicking. The reason for this outcome could be the potential discrepancies between strength production performance on the isokinetic

dynamometer and specific kicking performance in soccer [58]. Another factor could be that players may not reach their maximal strengths during ball kicking. In both cases, it appears that players exhibit different force–time profiles during isokinetic dynamometry compared to shot performance. While shot performance involves complete body biomechanics [15,46], players focusing on maximal force production and adopting the appropriate position during isokinetic performance may not align with the biomechanics of kicking. Therefore, although there might be some similarities between the EXT movement on the isokinetic dynamometer and shot performance in terms of biomechanical aspects, the moment of delivering the kick and the moment of producing maximal strength may not coincide. This outcome underscores the need for employing more specific methods tailored to soccer-specific force and timing investigations.

The findings of this research study generally indicate that concentric exercises designed at slow, moderate, and high angular velocities could enhance shot performance. Furthermore, the obtained results underscore the importance of giving equal consideration to the non-dominant extremity. However, certain precautions and additional tests are necessary to achieve more definitive conclusions.

The primary limitation of this study lies in the fact that strength data were solely acquired using the isokinetic dynamometer and directly correlated with ball speed. While the results provide a foundational understanding, obtaining clearer outcomes and establishing a more precise association between shot performance and isokinetic strength performance necessitates additional measures and supplementary tests. To this end, examining electromyographic (EMG) recordings from both performances can yield more precise data. This way, muscle contractions or contraction timings during both the shot and isokinetic strength measurement can be calculated, offering a more comprehensive insight into the relationship between the two. Furthermore, another limitation of the present study is the relatively small sample size (consisting of only 13 players). It is essential to include participants from different leagues and skill levels in future studies to enhance the generalizability of the research findings.

### 5. Conclusions

In conclusion, this research study has investigated the relationship between shot velocity and isokinetic strength parameters in soccer players. From the research study, it can be concluded that isokinetic torques can predict shot performance. It was observed that the extension strengths generated during concentric contractions positively influenced shot velocity. This finding underscores the significance of enhancing concentric strength to improve soccer players' shot performance.

We observed that developing balanced strength in the non-dominant extremity positively influenced shot performance across all angular velocities. This finding emphasizes the importance of the non-dominant extremity in soccer, highlighting that it can be trained using different exercise types if needed. Moreover, this result underscores the significance of non-dominant extremity performance in transferring force to the ball. However, considering the dynamic nature of modern soccer and the potential use of both extremities for shooting or maintaining balance, it can be stated that preparing both extremities adequately for various scenarios and planning sufficient strength and balance training is crucial. The study's findings suggest that strengthening both extremities in a balanced manner can aid soccer players in maintaining equilibrium and producing explosive movements during shots. However, a significant relationship was not observed between isometric and eccentric contractions' generated force and shot velocities. This indicates that there are differences between soccer players' isokinetic strength performance and their force performance during shooting.

These findings underscore the importance of implementing strategies to enhance concentric strength and the balance provided by the non-dominant extremity in improving soccer players' shot performance. Furthermore, delving deeper into the disparities between

isokinetic strength performance and actual shot performance in soccer players could contribute to more effective and player-focused training programs in the future.

Understanding the impact of various physical parameters on ball velocity among soccer players can provide valuable insights for developing customized training and nutrition programs to enhance performance. This study represents an essential step toward improving female soccer players' performance and better comprehending isokinetic force models associated with shot velocity. The findings offer valuable guidance for soccer coaches, athlete consultants, and sports scientists engaged in soccer training and performance enhancement.

**Author Contributions:** C.Ö., formulation of an idea or hypothesis for research, data collection, analysis, article writing, design, literature review; N.H., formulation of an idea or hypothesis for research, article writing, literature review; B.Y., formulation of an idea or hypothesis for research, data collection; S.H., article writing, critical review; P.F., article writing, critical review; A.S., article writing, critical review; M.A.K., article writing, critical review; A.İ., formulation of an idea or hypothesis for research, supervision and consulting, critical review. All authors have read and agreed to the published version of the manuscript.

**Funding:** This research received no external funding.

**Institutional Review Board Statement:** The study was conducted in accordance with the Declaration of Helsinki and approved by the Institutional Ethics Committee of ORDU UNIVERSİTY (protocol code 2023-137).

**Informed Consent Statement:** Informed consent was obtained from all subjects involved in the study. Written informed consent has been obtained from the patient(s) to publish this paper.

**Data Availability Statement:** Data is unavailable due to privacy restrictions.

**Conflicts of Interest:** The authors declare no conflict of interest.

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
