# Peer review of "Examining the Link between Isokinetic Strength Metrics and Ball Speed in Women’s Soccer"

_applsci, doi:10.3390/app132212217_

Round 1

Reviewer 1 Report

Comments and Suggestions for Authors

The association with ball speed is not only muscle strength, why did you focus on only muscle strength in particular knee muscle? Please provide the factors that relate to ball speed such as agility in soccer players. Again, muscle strength is not only knee muscles why did you focus on these muscles? 

Line 94: a large effect size was calculated by using G*power. Why did you use the large effect size? 

Line 329; please provide the reference. 

Comments on the Quality of English Language

 The manuscript needs some more clarification.

The writing style and language need to be improved.

Author Response

REVIEWER 1:

- The association with ball speed is not only muscle strength, why did you focus on only muscle strength in particular knee muscle? Please provide the factors that relate to ball speed such as agility in soccer players. Again, muscle strength is not only knee muscles why did you focus on these muscles? 

R: While the test is commonly referred to as measuring knee strength in most of the literature, it assesses the strength of the quadriceps and hamstring muscles during flexion and extension movements. In this research, the focus was on the performance of these muscles. Of course, during a shot, the entire body biomechanics are a crucial factor in determining shot performance. However, the relationships between the performance of the body's extremities and ball kinematics should be examined in detail. This way, the strength or power-related aspects of training can be effectively tailored.

The research identified performance differences even among angular velocities. These differences will provide valuable insights for coaches in structuring training. Furthermore, it is well known that isometric and eccentric contractions are significant kinetic factors in body biomechanics as they allow for the release of more torque compared to concentric contractions. However, when examined in the context of soccer, it was found that concentric contractions are the dominant contraction type. All of these findings will guide coaches and athletes in designing training related to leg strength. Therefore, this research focused solely on the lower extremities.

- Line 94: a large effect size was calculated by using G*power. Why did you use the large effect size?                                    

R: We were only able to reach our accessible sample group through large effect power analysis, and we could substantiate it with the specified statistics. To ensure the reliability of the results, we worked exclusively with top-level players, resulting in a limited sample size.

- Line 329; please provide the reference. 

R: The reference added to line 329. Thank you.

Reviewer 2 Report

Comments and Suggestions for Authors

I recommend redoing the design of the conclusions so that it includes obvious information about: 1. Row 66,67; 2, Row 72,73; 3. Row 85, 86, 87. We can also waive recommendations like the one in line 377...for future studies! The conclusions are for this study! In the conclusions, we should not find deliberately, conditional situations determined by the word IF (line 380)!

Author Response

- I recommend redoing the design of the conclusions so that it includes obvious information about: 1. Row 66,67; 2, Row 72,73; 3. Row 85, 86, 87. We can also waive recommendations like the one in line 377...for future studies! The conclusions are for this study! In the conclusions, we should not find deliberately, conditional situations determined by the word IF (line 380)!

R: The Conclusions section has been revised according to the recommendations.

Reviewer 3 Report

Comments and Suggestions for Authors

The topic is original and of significant scientific interest with interdisciplinary characteristics able to bring an advance with respect to the current state of the art present in the literature.

The conception, planning and organization of the text are rigorous and well structured

The text is characterized by good cohesion and clarity of exposition, supported by fully coherent and original arguments. The methodological choices are clear and straightforward.

The analysis of the literature is based on an in-depth and researched bibliographic research that includes all the most recent and important contributions in the literature.

Main concern:

1. Abstract. I suggest the authors to start with a brief intro that better highlights their work.

2. Discussion should be enriched with the existing theory.

Author Response

1. The topic is original and of significant scientific interest with interdisciplinary characteristics able to bring an advance with respect to the current state of the art present in the literature.

The conception, planning and organization of the text are rigorous and well structured.

The text is characterized by good cohesion and clarity of exposition, supported by fully coherent and original arguments. The methodological choices are clear and straightforward.

The analysis of the literature is based on an in-depth and researched bibliographic research that includes all the most recent and important contributions in the literature.

R: Thank you for your valuable comments

2. Abstract. I suggest the authors to start with a brief intro that better highlights their work.

R: A short introduction was added to the abstract.

3. Discussion should be enriched with the existing theory.

R: Some minor updates were made and checked in the Discussion section in line with the suggestions of reviewers.

Round 2

Reviewer 1 Report

Comments and Suggestions for Authors

The revised manuscript can be accepted. 

Comments on the Quality of English Language

English is fine and clear.